# Whey Protein Hydrolysate Exerts Anti-Inflammatory Effects to Alleviate Dextran Sodium Sulfate (DSS)-Induced Colitis via Microbiome Restoration

**DOI:** 10.3390/nu15204393

**Published:** 2023-10-17

**Authors:** Wenrong Zou, Zixin Fu, Xiaohong Guo, Lei Yao, Hui Hong, Yongkang Luo, Yuqing Tan

**Affiliations:** 1Key Laboratory of Functional Dairy, College of Food Science and Nutritional Engineering, China Agricultural University, Beijing 100083, China; sy20213061223@cau.edu.cn (W.Z.); fuzixin1998@126.com (Z.F.); hhong@cau.edu.cn (H.H.); luoyongkang@cau.edu.cn (Y.L.); 2Department of Product and Development, Hebei Dongkang Dairy Co., Ltd., Shijiazhuang 052165, China; guoxiaohong@bj-milkyway.com (X.G.); 18910005160@163.com (L.Y.)

**Keywords:** whey protein hydrolysate, colitis, gut microbiome, MHC class I pathway

## Abstract

Whey protein hydrolysate (WPH) has been shown to have a variety of bioactivities. This study aimed to investigate the preventive effect of WPH on dextran sodium sulfate (DSS)-induced colitis in C57BL/6J mice. The results indicated that WPH intervention for 37 days was effective in delaying the development of colonic inflammation, and high doses of WPH significantly inhibited weight loss (9.16%, *n* = 8, *p* < 0.05), protected the colonic mucosal layer, and significantly reduced the levels of inflammatory factors TNF-α, IL-6, and IL-1β in mice with colitis (*n* = 8, *p* < 0.05). In addition, WPH intervention was able to up-regulate the short-chain fatty acids secretion and restore the gut microbiome imbalance in mice with colitis. Notably, high-dose WPH intervention increased the relative abundance of *norank_f_Muribaculaceae* by 1.52-fold and decreased the relative abundance of *Romboutsia* and *Enterobacter* by 3.77-fold and 2.45-fold, respectively, compared with the Model group. WPH intervention protected colitis mice mainly by reversing the microbiome imbalance and regulating the major histocompatibility complex (MHC) class I pathway. This study showed that WPH has anti-inflammatory activity and a promising colitis management future.

## 1. Introduction

Ulcerative colitis (UC) is a chronic inflammatory bowel disease with the main symptoms of diarrhea and bloody stools. The pathogenesis of UC is still unclear, but studies have shown that genetic factors, environmental factors, immune dysregulation, and defects in the mucosal barrier and epithelial barrier are closely associated with the pathogenesis of UC [1]. UC is highly prevalent and difficult to cure and is often accompanied by serious complications, such as liver injury and lung injury, which seriously affect people’s health [2,3]. The WHO states that the prevalence of UC is about 0.20–0.25% and has classified UC as one of the clinically intractable diseases [4]. According to statistics, the incidence of UC is still on the rise worldwide [5]. Depending on the severity of the colitis disease, there are multiple treatment options for UC patients, including primarily multiple medications and surgery, but it is difficult to have a definitive treatment plan, primarily due to the advantages and disadvantages of each of the long-term effects of medication and surgical treatment [6]. Currently, the main medicines used to treat UC are salicylates, corticosteroids, and immunomodulators, but they can only relieve the symptoms, and patients usually need to take the medicines for a long time, which often has significant side effects, such as osteoporosis [7]. Therefore, there is an urgent need to find some safe and effective functional factors as alternatives to drugs for the treatment of colitis.

A bioactive peptide is a peptide composed of 2–20 amino acid residues. It has a wide range of functional activities, which, combined with its ease of metabolism and lack of side effects, is considered a good alternative to some synthetic drugs [8]. In recent years, more and more research has focused on bioactive peptides and found that they have a very promising future in the prevention and treatment of human diseases [9]. In particular, some studies have shown that bioactive peptides have a protective effect on colitis. Xuanwei ham-derived peptides could modulate inflammatory cell factors, protect the intestinal barrier, and regulate intestinal microbiota, thus alleviating DSS-induced intestinal inflammation in mice [10]. Trichotomy matsutake-derived peptides alleviated intestinal inflammation in DSS-induced colitis mice via the NF-κB/MLCK/p-MLC pathway [11]. Peptides from fresh wheat germ–apple fermentation were able to exert anti-colitis activity by regulating the MAPK pathway, alleviating the reduction in tight junction proteins, and promoting cell proliferation [12].

Whey protein is the by-product of dairy processing and is ubiquitously applied in food systems. The consumption of whey protein has a positive impact on intestinal health, as it regulates the immune function of the intestine and has a preventive effect on colonic damage caused by acetic acid [13]. It is worth noting that whey protein can be broken down into some small peptides and amino acids by enzymatic digestion and heat treatment to obtain whey protein hydrolysate [14]. Whey protein hydrolysate (WPH) has a variety of functional activities, is more easily absorbed by the body, and is often referred to as whey protein peptide. Given the positive effects of whey protein in mice with colitis, we speculate that WPH may also be a potential treatment for patients with UC. It has been shown that whey protein hydrolysate has various functional activities in anti-atherosclerosis and hepatoprotection [15], improvement of sarcopenia [16], radiation protection [17], antioxidant [18], and exhibits some anti-inflammatory activity while exerting these activities. However, there is a lack of mechanistic studies that address how WPH exerts anti-inflammatory activity. Therefore, the present study focused on the role of WPH in inflammatory diseases and aimed to elucidate the protective effect of WPH on DSS-induced colitis in mice and the potential mechanisms underlying its action to provide new ideas for the development and utilization of novel functional ingredients for the management of colitis.

## 2. Materials and Methods

### 2.1. Reagents and Chemicals

The WPH (HilmarTM 8350) used was a whey protein hydrolysate donated by Tianjin Milkway Import and Export Corp. (Tianjin, China) with 82% protein by dry weight; composition of whey protein hydrolysate was elucidated in Appendix A. DSS with a molecular weight of 36–50 KDa was purchased from MP Biomedicals (Santa Ana, CA, USA). The Fecal Occult Blood Test Kit was purchased from Zhuhai Biotechnology Co., Ltd. (Guangzhou, Guangdong, China). Mouse tumor necrosis factor α (TNF-α), interleukin 6 (IL-6), interleukin 1β (IL-1β), and lipopolysaccharides (LPS) ELISA Kits were purchased from Beijing BoRuiChangYuan Technology Co., Ltd. (Beijing, China).

### 2.2. Determination of Peptide Sequences in WPH

The peptide sequence of WPH was performed by liquid chromatography–tandem mass spectrometry (LC-MS/MS). The obtained data were retrieved from UniProt Knowledgebase (UniProtKB) database and PEAKS studio Version X+ (Bioinformatics Solutions Inc., Waterloo, Canada). Intensity value profiles of the peptides were visualized with online Peptigram.

### 2.3. Animal Experimental Design

Animal experimental design and husbandry were performed in strict accordance with the Beijing Regulations on the Management of Laboratory Animals and were approved by the Experimental Animal and Experimental Ethics Review Committee of China Agricultural University with license No. AW11702202-4-2. Healthy SPF-grade male C57BL/6J mice (6–8 weeks old; animal license No. SCXK(Jing)2021-0006; Certification No. 110011220107869312) were purchased from Beijing Vital River Laboratory Animal Technology Co., Ltd. (Beijing, China). C57BL/6J mice were housed in SPF-level barriers (temperature 24 ± 1 °C and 12 h cycle each for light and dark) and provided with a standard diet (SPF Biotechnology Co., Ltd., Beijing, China, Permit number: SCXK (Jing) 2019-0010) in the Experimental Animal Center of China Agricultural University West Campus.

After one week of acclimatization feeding, all animals were randomly divided into four groups (12 mice per group): normal control group (Control, gavage with saline), DSS model group (Model, gavage with saline), WPH low-dose group (WPH-L, gavage with 300 mg/kg·bw WPH), WPH high-dose group (WPH-H, gavage with 600 mg/kg·bw WPH). During the 38-day experimental period, saline was gavaged daily in the Control and Model groups, and WPH was gavaged daily in the WPH-L and WPH-H groups. All mice were fed water ad libitum, and 3% DSS drinking water was given to the Model, WPH-L, and WPH-H groups on days 31–37. The mice were euthanized at the end of this experiment, and blood, colon, colon contents, and other tissues were collected and stored in a refrigerator at −80 °C for subsequent analysis.

### 2.4. Disease Activity Index (DAI) Analysis

The body weight, fecal traits, and fecal blood were observed and recorded daily during the modeling period on days 31–37, and the DAI was calculated according to Appendix A to assess the severity of colitis in mice. The DAI score is the mean of the weight loss rate score, fecal traits score, and fecal occult blood score. The rate of weight loss is the percentage of daily weight after induction compared to the weight on the day before induction.

### 2.5. Colonic Length and Organ Index Analysis

The mice were dissected, the colon was taken out, and the length was measured; the liver, spleen, kidney, and thymus were taken out and weighed, and the organ index (organ weight/mouse body weight × 100%) was calculated.

### 2.6. Colon Histological Analysis

The distal colon was taken 1 cm, flushed with pre-chilled PBS, and fixed in 4% tissue fixative for 48 h. Sections (5 μm) were embedded in paraffin, and then colonic tissue damage was assessed with hematoxylin and eosin staining (H and E), and changes in colonic mucus were observed with Alisin blue-periodic acid Schiff’s stain (AB-PAS).

### 2.7. Analysis of Inflammatory Indicators and LPS in Serum

The collected blood was allowed to stand for 30 min and then centrifuged at 1180× *g* for 15 min, and the supernatant was taken for use. The levels of TNF-α, IL-6, IL-1β, and LPS in mouse serum were measured in strict accordance with the ELISA kit instructions.

### 2.8. Short-Chain Fatty Acid (SCFAs) Analysis

Standard curves were plotted by gas chromatography mass spectrometer (GC-MS) analysis of acetic acid, propionic acid, isobutyric acid, butyric acid, isovaleric acid, and valeric acid standards. The mouse feces were weighed, added to 10 times the volume (mass: volume = 1:10) of 3 M sulfuric acid solution, sonicated for 40 min, and then centrifuged, and the supernatant was added to a certain amount of internal standard (volume of supernatant: volume of internal standard = 10:1) and analyzed by GC-MS. The internal standard was obtained by dissolving 2-methylpentanoic acid in water.

### 2.9. Proteomic Analysis of Colonic Tissue

#### 2.9.1. Extraction of Total Colonic Protein

Colon samples were taken and ground into powder in liquid nitrogen, then added to protein lysis solution (100 mM ammonium bicarbonate, 8 M urea) for ultrasonic lysis in an ice bath for 5 min and then centrifuged at 4 °C and 12,000× *g* for 15 min; the supernatant was added with DTT, IAM, and acetone and centrifuged after the reaction, and the precipitate was collected.

#### 2.9.2. TMT Labeling

The extracted protein precipitate was solubilized by adding a proteolytic solution (8 M urea, 100 mM TEAB). Then trypsin was added at 37 °C for 4 h, and then trypsin and CaCl_2_ were added to continue the digestion overnight. Next, pH < 3 was adjusted with formic acid, and after centrifugation (12,000× *g* for 5 min), the supernatant was desalted on a C18 column and dried under a vacuum. Then 0.1 M TEAB was added for re-solubilization, followed by acetonitrile solubilized TMT labeling reagent for 2 h. The reaction was finally terminated by adding ammonia, desalted, and vacuum drying.

#### 2.9.3. LC-MS/MS Analysis

The lyophilized powder was dissolved with mobile phase A (2% acetonitrile, 98% water) and centrifuged at 12,000× *g* for 10 min. Then the gradient elution was performed by HPLC on a Waters BEH C18 (4.6 × 250 mm, 5 μm), and the column temperature was set at 45 °C. Immediately afterward, the separated fractions were dried under vacuum and then re-solubilized with 0.1% formic acid. The fractions were eluted by liquid chromatography using an EASY-nLCTM 1200 nano-upgrade UHPLC system. MS/MS analysis was then performed with a Q ExactiveTM series mass spectrometer. The mass spectra were acquired in a data-dependent acquisition mode with a full scan range of m/z 350–1500. The mass spectral data were processed using Proteome Discoverer 2.5 (Thermo Scientific, Tewksbury, MA, USA) software. The mass tolerances of precursor and fragment ions were set to 10 ppm and 0.02 Da, respectively. And the maximum number of missed cut sites allowed was two.

#### 2.9.4. Bioinformatics Analysis

The protein database used in this study was Mus_musculus_uniprot_2022_9_5.fasta.fasta (86,436 sequences). And screening for differentially expressed proteins (DEPs) by fold change was >1.2 or < 0.83, *p*-value < 0.05. Next, GO (http://geneontology.org/ accessed on 25 September 2023) enrichment analysis and KEGG pathway (http://www.genome.jp/kegg/ accessed on 25 September 2023) analysis were performed with the NovoCloud platform (https://magic.novogene.com accessed on 25 September 2023).

### 2.10. Gut Microbiome Analysis

Microbial community genomic DNA from colon contents was extracted using the E.Z.N.A.^®^ Soil DNA Kit (Omega Bio-Tek, Norcross, GA, USA). The concentration and purity of extracted DNA were determined by 1% agarose gel electrophoresis and NanoDrop^®^ ND-2000 (Thermo Scientific Inc., St. Louis, MA, USA). PCR amplification of the V3-V4 region in 16S rRNA was performed using primers 338F (5′-ACTCCTACGGGAGGCAGCAG-3′) and 806R (5′-GGACTACHVGGGTWTCTAAT-3′). The reaction system consisted of 4 μL 5 × TransStart FastPfu buffer, 2 μL 2.5 mM dNTPs, 0.8 μL 338F (5 μM), 0.8 μL 806R (5 μM), 0.4 μL TransStart FastPfu DNA polymerase, and 10 ng template DNA in a final volume of 20 μL. The PCR products were recovered using a 2% agarose gel and purified using the AxyPrep DNA Gel Extraction Kit (Axygen Bioscience, Union City, CA, USA) and quantified using QuantusTM Fluorometer (Promega, Madison, WI, USA), then sequenced using the Illumina MiSeq PE300 platform (Illumina, San Diego, CA, USA) according to the standard protocols by Majorbio Bio-Pharm Technology Co. Ltd. (Shanghai, China). The optimized sequences were then clustered into operational classification units (OTUs) using UPARSE 7.1 with a sequence similarity of 97%. OTU taxonomy was annotated using RDP Classifier 2.2 against the Silva 16S rRNA database (v138). The raw data were deposited into the NCBI Sequence Read Archive (SRA) database with accession number PRJNA958610.

### 2.11. Data Analysis

More than 6 mice in each group were randomly selected for different data measurements. The experimental data were expressed as mean ± standard deviation. Statistical analysis and graphing were performed using GraphPad 9.4.1 software. Significance analysis was performed using ANOVA and Tukey’s multiple comparisons, and data were considered significantly different when *p* < 0.05.

## 3. Results and Discussion

### 3.1. The Composition of WPH

As shown in Appendix A, WPH consists of albumin (28.35%), β-lactoglobulin (21.46%), lactotransferrin (18.58%), α-lactoglobulin (16.48%), polymeric immunoglobulin receptor (11.11%), and folate receptor α (4.02%). The best unique peptide–spectrum match (PSM) in WPH is DSPDLPKLKPDPNTL (sequence residues No. 132–146), derived from albumin (Appendix A). Albumin has been shown to have various functions such as anti-inflammatory and antioxidant, and it can modulate the immune response by binding to bacterial antigens and LPS as well as inhibiting inflammatory cell adhesion [19]. And WPH contains a high percentage of albumin; thus, we hypothesize that this may be an important reason for the anti-inflammatory active component of WPH.

### 3.2. Effect of WPH Intervention on Body Weight and DAI Score in Mice with DSS-Induced Colitis

The symptoms of DSS-induced colitis in mice are very similar to those of human UC. To investigate the protective effect of WPH on colitis, 3% DSS was used to induce C57BL/6J mice to establish an acute colitis model, and the specific experimental protocol is shown in Figure 1A. Common clinical signs in patients with colitis include weight loss and bloody stools visible to the naked eye, and similar symptoms were observed in mice with colitis in this experiment [20]. The changes in the body weight of the mice in each group during DSS induction are shown in Figure 1B. Compared with the Control group, the body weight of the Model, WPH-L, and WPH-H groups induced by DSS were significantly reduced (*p* < 0.05). At the same time, the high dose of WPH intervention significantly inhibited the reduction in the body weight of the mice compared with the Model group (*p* < 0.05), and the WPH-L group also delayed the reduction in the body weight of the mice with colitis to some extent. The DAI score, a composite score of the rate of weight loss, fecal traits, and blood in stool, is an essential indicator of the success of modeling colitis in mice. As shown in Figure 1C, compared with the Control group, the DAI scores were significantly higher in the Model, WPH-L, and WPH-H groups on days 2, 3, and 4 of DSS induction, respectively (*p* < 0.05). At the end of DSS induction, the DAI score was significantly higher in the Model group compared to the Control group (*p* < 0.05); the DAI score was considerably lower in the WPH-H group compared to the Model group (*p* < 0.05). WPH intervention slowed down weight loss and reduced the DAI score. This suggests that WPH intervention can alleviate the development of colitis to some extent and that high doses of WPH have better results in maintaining the weight of patients with colitis.

### 3.3. Effect of WPH Intervention on Colon and Organ Index in Mice with DSS-Induced Colitis

Usually, colitis mice are accompanied by a shortening of the colon length, which is mainly caused by swelling and congestion of the colon. The changes in the colonic length of the mice in different treatment groups are shown in Figure 2A,B. Compared with the Control group, the colon length in the Model group was significantly shortened after DSS induction (*p* < 0.05), while the colon length in the WPH-L and WPH-H groups treated with WPH was significantly increased compared with the Model group (*p* < 0.05). Next, the pathological damage of the colon and the integrity of the mucus layer were assessed by HE staining and AB-PAS staining, and the results (Figure 2D,E) showed that the colonic crypt and glandular structures of mice in the Control group were intact, and the colonic mucus layer was intact and evenly distributed and contained a large number of goblet cells covered with mucus without obvious inflammatory cell infiltration. After DSS induction, the colonic crypt structure of the mice in the Model group was severely damaged or even disappeared, the mucus layer was severely damaged or even invisible, and the number of goblet cells was also greatly reduced with severe edema of the submucosa, accompanied by inflammatory cell infiltration. In contrast, WPH intervention significantly alleviated the pathological symptoms of the colon and improved the massive reduction in goblet cells and mucus in the mice with colitis; especially, the colonic crypt structure of the mice in the WPH-H group was more intact, with the presence of a large number of goblet cells and a few inflammatory cells. It has been shown that the number of cupped cells is related to the severity of colonic inflammation, and the considerable reduction in cupped cells in UC patients leads to a decrease in mucus secretion, resulting in a defective mucus layer [21]. The results of mucus area quantification (Figure 2C) showed that low and high doses of WPH significantly increased the mucus area by 54.02% (*p* < 0.001) and 61.50% (*p* < 0.0001), respectively. Ma et al. [22] found that Propionibacterium fowler could alleviate colitis by increasing the number of cupped cells and stimulating their mucus secretion, which is consistent with the results of WPH intervention in this study. This suggests that WPH intervention reduces histopathological damage in the colon and can have the effect of increasing the thickness of the mucus layer by increasing the number of cupped cells, which protects the intestinal epithelial barrier in mice with colitis, thereby providing relief from colitis.

Table 1 shows significant changes in the thymus, spleen, liver, and kidney organ indices in the Model group compared to the Control group (*p* < 0.05). The WPH intervention significantly reversed changes in the thymus and liver (*p* < 0.05) and reduced changes in the spleen and kidneys to some extent. The thymus and spleen are the immune organs of the body and play an important role in the body’s inflammatory response. WPH intervention suppressed thymic atrophy and splenomegaly caused by colitis, which is consistent with the findings of Cao and Lv et al. [23,24]. These results suggest that WPH has immunomodulatory potential in colitis.

### 3.4. Effect of WPH Intervention on Serum Inflammatory Factors in DSS-Induced Colitis Mice

Overexpression of some pro-inflammatory factors, such as TNF-α, IL-6, and IL-1β, can cause damage to the intestinal epithelial barrier and promote the development of colitis [25]. Therefore, we measured the levels of inflammatory factors in the serum of mice in different groups, and the results are shown in Figure 3A–C. The levels of pro-inflammatory factors TNF-α, IL-6, and IL-1β were significantly increased in the Model group with the induction of DSS compared with the Control group (*p* < 0.05). At the same time, the WPH intervention significantly inhibited the increase in the levels of these pro-inflammatory factors (*p* < 0.05). Still, there was no significant difference between the low and high doses. Intestinal tissues of UC patients are often accompanied by an overexpression of inflammatory factors such as TNF-α, IL-6, and IL-1β, and the expression levels of these pro-inflammatory factors are positively correlated with the severity of UC [26]. This is consistent with the results of the present study in which the Model group of mice suffered from the most severe colitis and had the highest expression levels of pro-inflammatory factors. The overexpression of pro-inflammatory factors can damage the intestinal epithelial barrier, leading to increased intestinal permeability, which in turn causes the transfer of some intestinal bacteria and intestinal dysfunction [27]. Thus, we hypothesized that WPH intervention could protect the intestinal barrier by inhibiting the elevated levels of pro-inflammatory factors, which could play a role in maintaining the balance of the gut microbiome.

### 3.5. The Gut Microbiome of DSS-Induced Colitis Mice

UC patients are often accompanied by an imbalance in the gut microbiome, leading to an overgrowth of harmful intestinal bacteria, which can directly damage the colonic epithelium due to the reduction in the colonic mucus layer, causing damage to the intestinal mucosal barrier and inducing a more severe inflammatory response [28,29]. Therefore, this study analyzed the effect of WPH intervention on the gut microbiome of mice with colitis by 16S rRNA sequencing. The results of alpha diversity (Figure 4A,B) showed that after DSS induction, Simpson and Chao’s indexes were reduced in the Model group. In contrast, WPH intervention did not have a significant inhibitory effect on the reduction in alpha diversity. The results of beta diversity (Figure 4C,D) showed that the gut microbiome composition of mice in the Model group was significantly separated from that of the Control group, the two groups were distant from each other, and the distribution of individuals in the Model group was scattered and different. A high dose of WPH intervention could effectively reduce the intra-group differences and make the distribution of individuals in the group more aggregated and move toward the Control group. This is consistent with the results of Nigella A improving the gut microbiome of mice with colitis [30]. WPH intervention may alleviate colitis by reducing the alteration of intestinal flora.

In addition, the present study analyzed the changes of WPH intervention on the gut microbiome composition of mice with colitis at the phylum level and genus level, and the results are shown in Figure 4E,F. Firmicutes, Bacteroidetes, Proteobacteria, and Actinobacteria in the Control, Model, WPH-L, and WPH-H groups accounted for more than 90% of the relative abundance of all clades. After DSS induction, the intestinal microorganisms in each group changed at the phylum level, mainly as follows: Compared with the Control group, the relative abundance of Bacteroidetes and Actinobacteria decreased and that of Proteobacteria increased in the Model group, while the WPH intervention was able to increase the relative abundance of Bacteroidetes and Actinobacteria and decreased the relative abundance of Proteobacteria. The genus-level clustering results of the first 20 genera (Figure 4F) showed that the relative abundance of *norank_f_Muribaculaceae* in the Model group significantly decreased to 11.12%, and the relative abundance of *Romboutsia* and *Enterobacter* significantly increased to 9.10% and 20.15% (*p* < 0.05) compared to the Control group. And WPH intervention was able to suppress the imbalance in these three genera; especially, high doses of WPH intervention increased the relative abundance of *norank_f_Muribaculaceae* by about 1.52-fold and decreased the relative abundance of *Romboutsia* and *Enterobacter* by about 3.77-fold and 2.45-fold, respectively, compared with the Model group. The abundance of *Romboutsia* increases in various diseases, such as intestinal stress syndrome and gastric cancer, and it has been shown to play a pro-inflammatory role in colitis [31]. *Enterobacter*, a pathogenic genus of Enterobacteriaceae, promotes the secretion of pro-inflammatory factors, disrupts intestinal mucosal tight junctions, increases intestinal permeability, and exacerbates inflammation [32]. In contrast, WPH intervention could inhibit the proliferation of these two harmful bacteria and reduce intestinal damage to some extent. In addition, DSS induction led to a decrease in the relative abundance of *norank_f_Muribaculaceae* in colitis mice, which is consistent with the study by Guo et al. [33]. It has been shown that *norank_f_Muribaculaceae* is a beneficial bacterium that can use intestinal mucus polysaccharides as growth nutrients to inhibit the growth and colonization of pathogenic bacteria in the intestine and improve intestinal mucosal damage by occupying its ecological space [34]. In the present study, WPH intervention reversed the changes in *norank_f_Muribaculaceae*. The above results suggest that WPH can promote beneficial bacteria and inhibit the proliferation of harmful bacteria, i.e., alleviate colitis by regulating the imbalance of intestinal flora.

### 3.6. Effect of WPH Intervention on LPS and SCFAs in Mice with DSS-Induced Colitis

LPS and SCFAs are the two main substances of gut microbial origin that play a facilitating (LPS) or alleviating (SCFAs) role in the development of colitis. Therefore, we analyzed serum LPS and fecal SCFA levels in four groups of mice, and the results are shown in Figure 5. Compared with the Control group, the serum LPS levels of the mice in the Model group were significantly higher after DSS induction (*p* < 0.05), and the levels of acetic acid, propionic acid, isobutyric acid, butyric acid, isovaleric acid, and valeric acid were significantly lower (*p* < 0.05). The low dose of WPH significantly inhibited the decrease in acetic acid, isobutyric acid, butyric acid, and valeric acid levels (*p* < 0.05), and the high dose of WPH significantly inhibited the increase in LPS levels and the decrease in propionic acid, isobutyric acid, butyric acid, isovaleric acid, and valeric acid levels (*p* < 0.05). SCFAs are important metabolites of intestinal microorganisms, which can not only positively influence human metabolism but also play an essential role in improving the damaged intestinal barrier and regulating the immune system of the body in patients with colitis [35,36]. Studies have shown that SCFAs can exert anti-inflammatory effects by regulating immune cell function and cytokine secretion and can also prevent colon carcinogenesis [37]. Conversely, LPS can exacerbate inflammatory responses and oxidative stress and affect intestinal barrier function [38]. This study found that WPH intervention could rescue the reduction in the level of SCFAs induced by DSS and increase the level of LPS, and the effect was more pronounced with high doses of WPH. This may be mainly because WPH intervention increased the relative abundance of SCFAs-producing Bacteroidetes and decreased the relative abundance of LPS-producing Proteobacteria [39,40]. Therefore, WPH can regulate the composition of the gut microbiome, which regulates the levels of intestinal metabolite SCFAs and LPS, which in turn regulate immune cell (e.g., antigen-presenting cells) function and cytokine (e.g., TNF-α, IL-6, and IL-1β in Figure 3) secretion. Overall, these findings emphasize the importance of WPH intervention in regulating the gut microbiota and its metabolites in the prevention of colitis, which may be an important pathway for WPH to prevent colitis, i.e., by regulating the composition of the gut microbial community to improve gut health and modulate the immune response, leading to the alleviation of colitis. However, the interactions between gut microbiota and immune response are complex, and the specific mechanism of action needs to be further explored.

### 3.7. Functional Annotation and Enrichment Analysis of Colonic Differential Proteins

To further investigate the mechanism by which WPH ameliorates colitis, the effect of WPH intervention on the colonic protein expression in the mice with colitis was clarified by the TMT protein quantification technique, and the related pathways were predicted. The volcano plot of the DEPs (Figure 6A,B) showed that 922 different proteins were identified in the Model group compared to the Control group (Model/Control), with 491 up-regulated proteins and 431 down-regulated proteins. A total of 62 differential proteins were identified in the WPH-H group compared to the Model group (WPH-H/Model), with 20 up-regulated proteins and 42 down-regulated proteins. Among all differential proteins, the protein that was down-regulated by Model/Control and up-regulated by WPH-H/Model was CA2; the protein that was up-regulated by Model/Control and down-regulated by WPH-H/Model was ITIH4. CA2 is carbonic anhydrase 2, a member of the carbonic anhydrase (CA) family, which is involved in various processes such as intra- and extracellular pH homeostasis and vascular regulation [41]. The down-regulation of CA2 protein in the colonic mucosa of UC patients causes an imbalance of colonic luminal pH and disruption of the mucosal epithelial barrier, which leads to uncontrolled inflammation in UC [42]. ITIH4 is an α-trypsin inter inhibitor heavy chain 4, associated with cell proliferation and migration during the acute phase inflammatory response, and ITIH4 may be an acute phase protein induced by IL-6 or LPS [43]. Thus, WPH can attenuate epithelial damage and reduce the inflammatory response by up-regulating CA2 protein, while the down-regulation of ITIH4 protein verifies that WPH can inhibit IL-6 and LPS production.

Next, we performed GO enrichment analysis of the DEPs, and the results (Figure 6C) showed that nine biological processes, four cellular components, and six molecular functions were annotated to the differential proteins in the WPH-H/Model groups. Among them, biological process analysis showed that most of the differential proteins were enriched in carbohydrate derivative metabolism and cellular component organization; according to cellular component analysis, most of the differential proteins were located in extracellular regions; molecular function analysis showed that most of the differential proteins were involved in receptor binding, transferring enzyme activity, and transferring glycosyl groups. In addition, we identified the most prominent biochemical metabolic and signaling pathways involved in differential proteins by KEGG pathway analysis. The results showed (Figure 6D) that the DEPs in the WPH-H/Model were mainly enriched in the antigen processing and presentation pathway, which is thought to be a possible critical pathway for the induction of colitis [44]. And the differentially expressed proteins induced by WPH intervention were mainly concentrated in the MHC class I pathway of the antigen processing and presentation pathway. In the presence of gut microbial dysbiosis, overstimulation of CD8 T cells promotes the development of inflammation and early T cell depletion, thereby reducing immunity [45]. And in mice with colitis, inflammatory tissues tend to be infiltrated by large numbers of activated NK cells (natural killer cells), which accumulate large amounts of pro-inflammatory factors and cause damage to the intestinal barrier [46,47]. Therefore, we speculate that gavage of high-dose WPH down-regulates the expression of differential proteins in the MHC class I pathway, reduces antigen presentation, decreases the activation of CD8 T cells and NK cells, and reduces the levels of pro-inflammatory factors, thereby reducing the disease severity of colitis. In other words, this study builds on the large body of research that has been performed on modulating inflammatory responses through antigen processing and delivery pathways to alleviate colitis and, more specifically, identifies WPH as acting through the MHC class I pathway of the antigen processing and delivery pathway.

The possible mechanisms of WPH intervention to alleviate colitis are shown in Figure 7. On the one hand, WPH reduces intestinal epithelial damage by regulating the balance of the gut microbiome. On the other hand, WPH may alleviate colitis by regulating the inflammatory response mainly through the MHC class I pathway.

## 4. Conclusions

This study showed that WPH has a preventive effect on 3% of DSS-induced colitis in C57BL/6J mice. WPH intervention alleviated disease activity in mice with colitis and protected the colonic mucus layer, which in turn maintained gut microbiota homeostasis and consequently promoted the production of SCFAs. In addition, WPH significantly inhibited the production of pro-inflammatory factors (TNF-α, IL-6, and IL-1β), thereby protecting the intestinal barrier in mice. The mechanism by which WPH prevents colitis may be primarily by regulating the gut microbiota and modulating the inflammatory response through the MHC class I pathway. Although high-dose WPH interventions seem to have better results in preventing colitis, the preventive effect caused by the level of the dose needs to be further investigated. The current study provides a new strategy for developing novel functional ingredients for colitis management.

## Figures and Tables

**Figure 1 nutrients-15-04393-f001:**
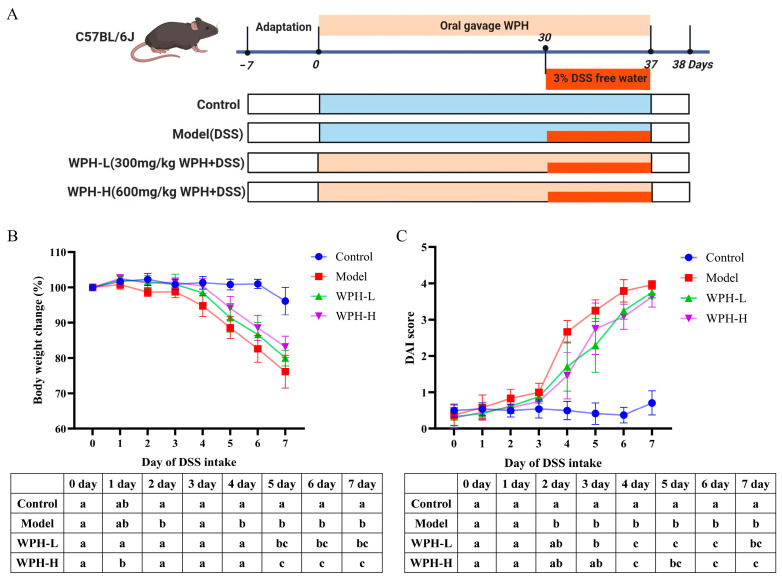
Effect of whey protein hydrolysate intervention on body weight, and DAI in mice with colitis. (**A**) Animal experimental design, (**B**) change in body weight, and (**C**) change in DAI. All outcome values are expressed as mean ± SD (*n* = 8). The statistical analysis was listed in the table below each figure, and different lowercase letters indicate a statistically significant difference (*p* < 0.05) between groups at the same time (**B**,**C**). WPH: whey protein hydrolysate (Hilmar 8350); Control: gavage saline + normal drinking water; Model: gavage saline + 3% DSS drinking water; WPH-L: gavage 300 mg/kg·bw WPH + 3% DSS drinking water; WPH-H: gavage 600 mg/kg·bw WPH + 3% DSS drinking water; DAI: disease activity index.

**Figure 2 nutrients-15-04393-f002:**
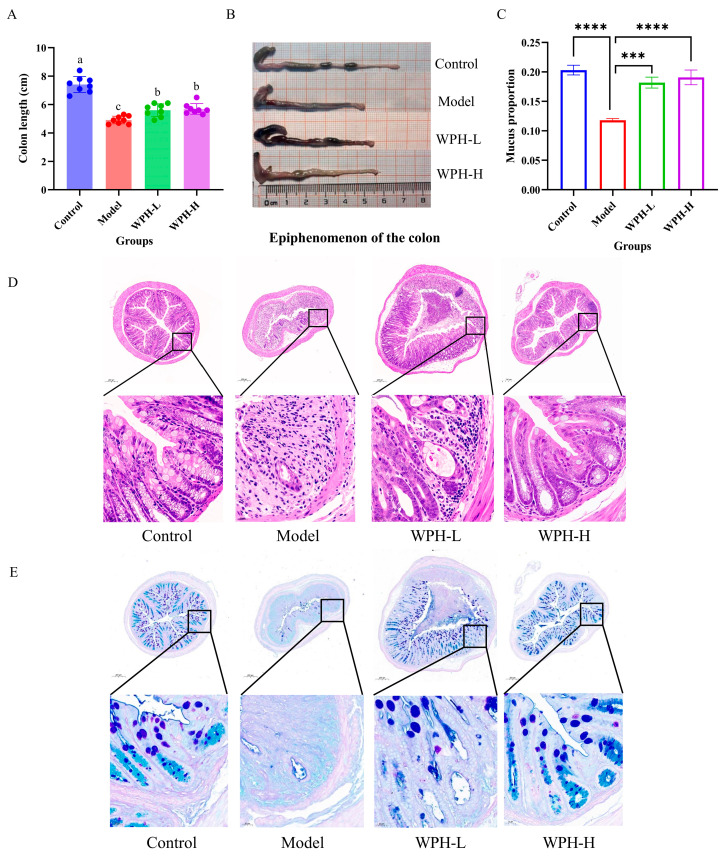
Effect of whey protein hydrolysate intervention on the length, histopathological symptoms, and mucus layer of the colon in mice with colitis. (**A**) Change in colonic length, (**B**) colonic appearance, (**C**) change in mucus area (mucus area/colon section area), (**D**) HE staining of the distal colon with magnifications of 200 μm (top) and 20 μm (bottom), and (**E**) AB-PAS staining of the distal colon with magnifications of 200 μm (top) and 20 μm (bottom). All resultant values are expressed as mean ± SD. Bars with different lowercase letters are significantly different between treatment groups (*p* < 0.05). *** and **** mean *p* < 0.001 and *p* < 0.0001. WPH: whey protein hydrolysate (Hilmar 8350); Control: gavage saline + normal drinking water; Model: gavage saline + 3% DSS drinking water; WPH-L: gavage 300 mg/kg·bw WPH + 3% DSS drinking water; WPH-H: gavage 600 mg/kg·bw WPH + 3% DSS drinking water.

**Figure 3 nutrients-15-04393-f003:**
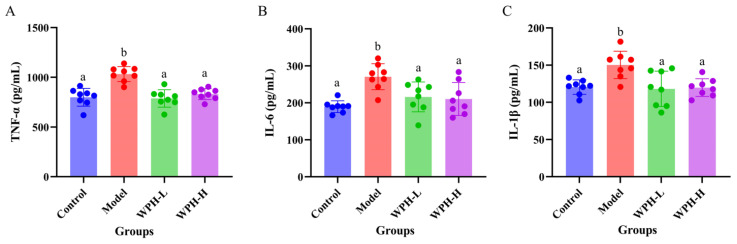
Effect of whey protein hydrolysate intervention on cytokines in mice with colitis. (**A**) TNF-α, (**B**) IL-6, and (**C**) IL-1β. All outcome values are expressed as mean ± SD (*n* = 8). Bars with different lowercase letters are significantly different (*p* < 0.05) between treatment groups. WPH: whey protein hydrolysate (Hilmar 8350); Control: gavage saline + normal drinking water; Model: gavage saline + 3% DSS drinking water; WPH-L: gavage 300 mg/kg·bw WPH + 3% DSS drinking water; WPH-H: gavage 600 mg/kg·bw WPH + 3% DSS drinking water.

**Figure 4 nutrients-15-04393-f004:**
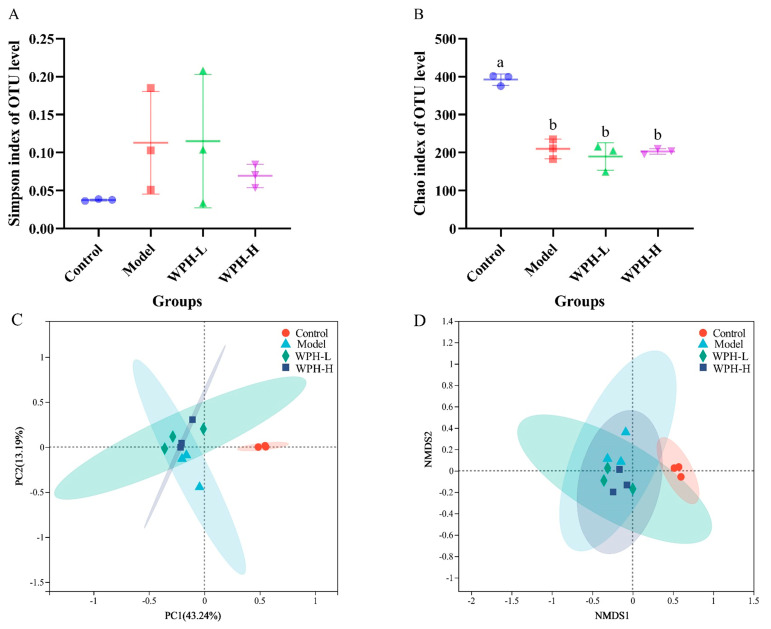
Effect of whey protein hydrolysate intervention on gut microbes in colonic contents of mice with colitis. (**A**) Simpson index, (**B**) Chao index, (**C**) PCoA, (**D**) NMDS, (**E**) phylum level, and (**F**) effect of relative abundance of the first 20 genera (compared to Control group). All resultant values are expressed as mean ± SD (*n* = 6). Bars with different lowercase letters are significantly different between treatment groups (*p* < 0.05). *, **, and **** mean *p* < 0.05, *p* < 0.01 and *p* < 0.0001. WPH: whey protein hydrolysate (Hilmar 8350); Control: gavage saline + normal drinking water; Model: gavage saline + 3% DSS drinking water; WPH-L: gavage 300 mg/kg·bw WPH + 3% DSS drinking water; WPH-H: gavage 600 mg/kg·bw WPH + 3% DSS drinking water.

**Figure 5 nutrients-15-04393-f005:**
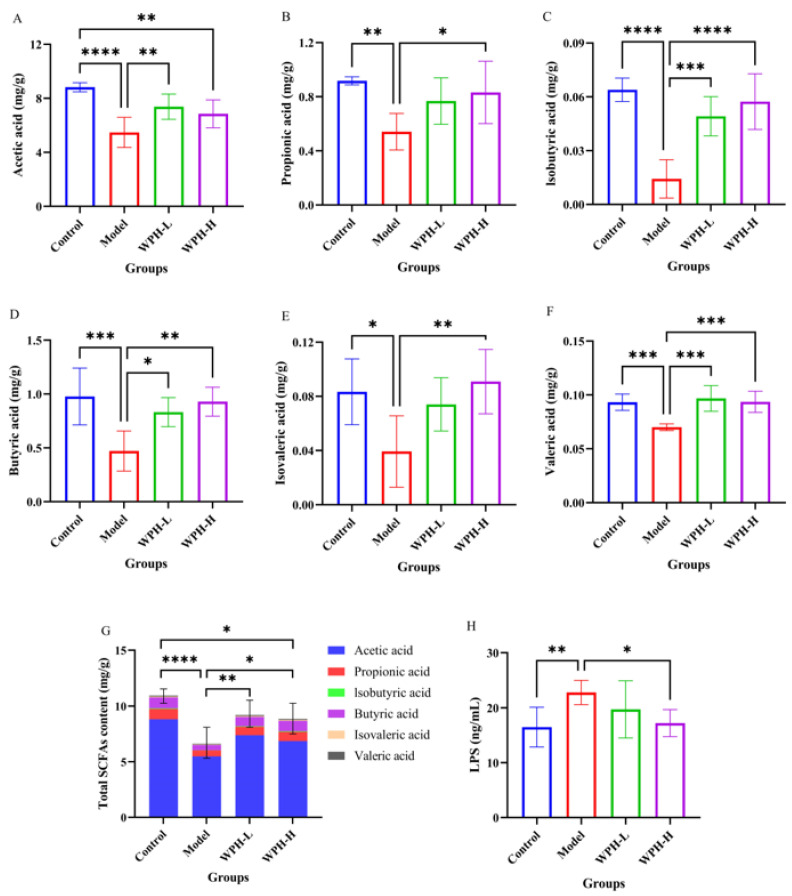
Effect of whey protein hydrolysate intervention on short-chain fatty acids and serum LPS in the feces of mice with 3% DSS-induced colitis. (**A**) Acetic acid, (**B**) propionic acid, (**C**) isobutyric acid, (**D**) butyric acid, (**E**) isovaleric acid, (**F**) valeric acid, (**G**) total short-chain fatty acid content, and (**H**) LPS levels in serum. All outcome values are expressed as mean ± SD (*n* = 6). *, **, ***, and **** mean *p* < 0.05, *p* < 0.01, *p* < 0.001, and *p* < 0.0001. WPH: whey protein hydrolysate (Hilmar 8350); Control: gavage saline + normal drinking water; Model: gavage saline + 3% DSS drinking water; WPH-L: gavage 300 mg/kg·bw WPH + 3% DSS drinking water; WPH-H: gavage 600 mg/kg·bw WPH + 3% DSS drinking water.

**Figure 6 nutrients-15-04393-f006:**
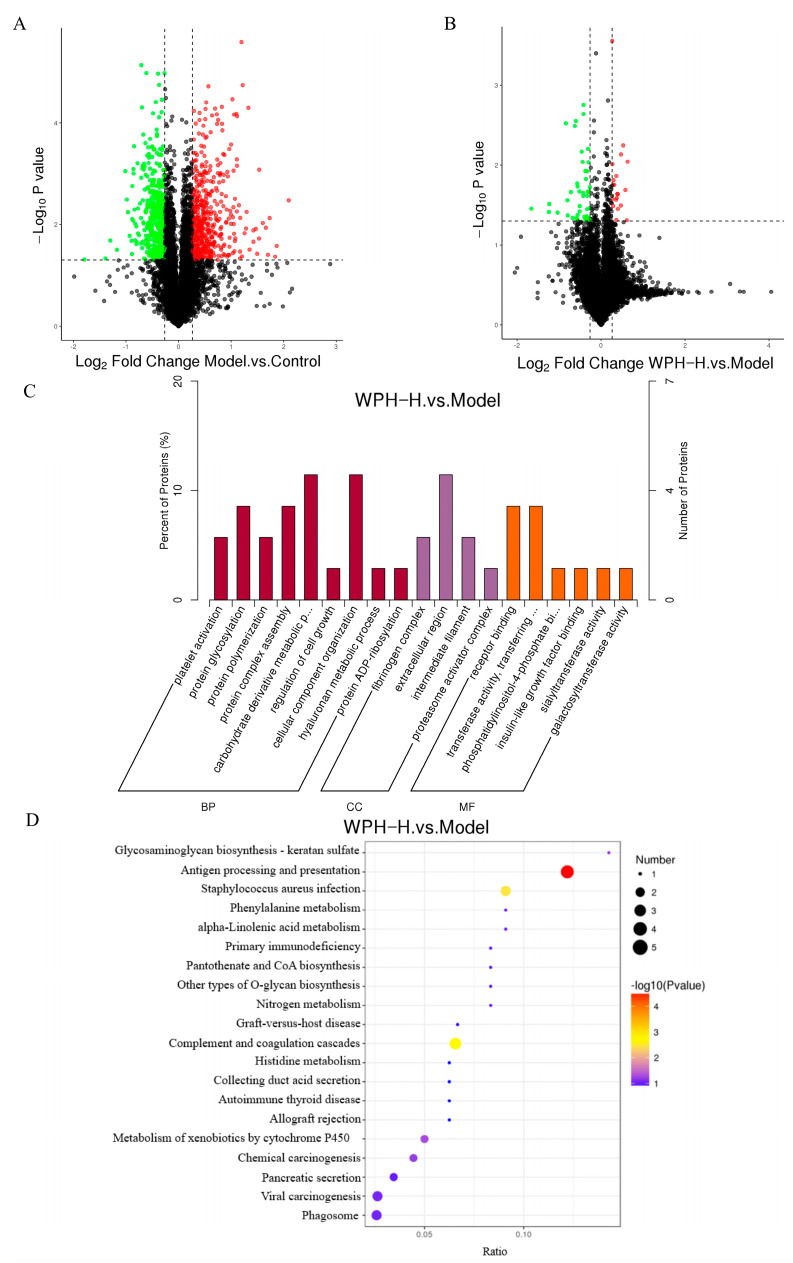
Proteomic analysis of the effect of WPH intervention on colonic tissue protein expression in mice with colitis. (**A**) Differential protein volcano plot of the Model group vs. the Control group, (**B**) differential protein volcano plot of the WPH-H group vs. the Model group, (**C**) GO annotation analysis of differential proteins (the WPH-H group vs. the Model group), and (**D**) KEGG pathway annotation analysis of differential proteins (the WPH-H group vs. the Model group). WPH: whey protein hydrolysate (Hilmar 8350); Control: gavage saline + normal drinking water; Model: gavage saline + 3% DSS drinking water; WPH-H: gavage 600 mg/kg·bw WPH + 3% DSS drinking water.

**Figure 7 nutrients-15-04393-f007:**
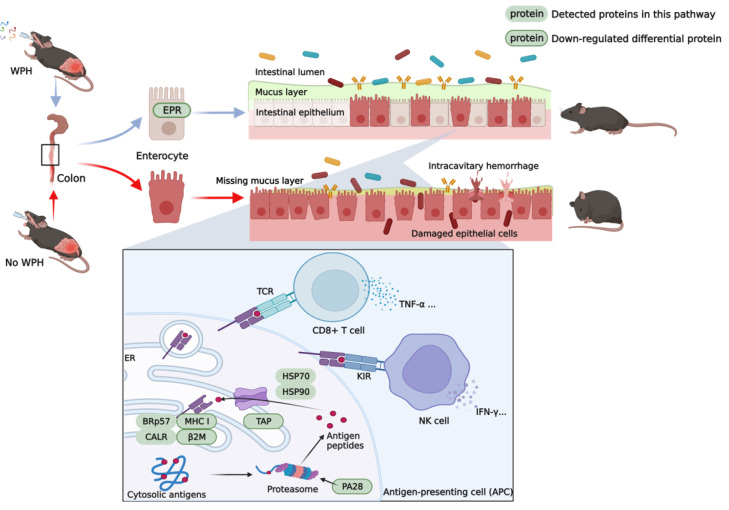
Mechanism of whey protein hydrolysate intervention to ameliorate 3% DSS-induced colitis in mice. WPH: whey protein peptide (Hilmar 8350); ER: endoplasmic reticulum; TCR: T cell receptor; TNF-α: tumor necrosis factor α; KIR: killer inhibitory receptor; NK cell: natural killer cells; IFN-γ: Interferon γ; BRp57, CALR, HSP70, HSP90: proteins detected in the MHC class I pathway; MHC I, β2m, TAP, PA28: down-regulated proteins detected in the MHC class I pathway.

**Table 1 nutrients-15-04393-t001:** Effect of whey protein hydrolysate intervention on the organ (thymus, liver, spleen, and kidney) ratio of 3% DSS-induced colitis mice.

	Thymus (%)	Liver (%)	Spleen (%)	Kidney (%)
Control	0.13 ± 0.01 a	3.81 ± 0.13 a	0.23 ± 0.01 a	1.22 ± 0.02 a
Model	0.06 ± 0.01 b	4.54 ± 0.15 b	0.39 ± 0.06 b	1.38 ± 0.08 b
WPH-L	0.09 ± 0.03 c	4.03 ± 0.24 a	0.35 ± 0.09 b	1.31 ± 0.04 b
WPH-H	0.10 ± 0.02 c	3.88 ± 0.35 a	0.36 ± 0.04 b	1.31 ± 0.05 b

All outcome values are expressed as mean ± SD (*n* = 8). Different lowercase letters are significantly different between treatment groups (*p* < 0.05). WPH: whey protein hydrolysate (Hilmar 8350); Control: gavage saline + normal drinking water; Model: gavage saline + 3% DSS drinking water; WPH-L: gavage 300 mg/kg·bw WPH + 3% DSS drinking water; WPH-H: gavage 600 mg/kg·bw WPH + 3% DSS drinking water.

## Data Availability

The data that support the findings of this study are available from the corresponding author upon reasonable request.

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
