# Peer review of "Whey Protein Hydrolysate Exerts Anti-Inflammatory Effects to Alleviate Dextran Sodium Sulfate (DSS)-Induced Colitis via Microbiome Restoration"

_nutrients, 2023, doi:10.3390/nu15204393_

Round 1

Reviewer 1 Report

The topic proposed by the researchers is attractive to researchers and the reading public. However, some information still needs to be clarified/corrected.

In the text, reference numbers should be placed in square brackets [ ], and placed before the punctuation; for example [1], [1–3] or [1,3] - apply for the entire text.

Ln 126 – Unless you report the average radius of the rotor, rpm is meaningless.  You should always report relative centrifugation force as “times g”.

The conclusion should be reformulated in the contest of the multitudes of results. Also, I recommend some conclusions regarding the relationship between WPH-L and WPH-H groups.

Reviewer 2 Report

The manuscript presented for the review addresses an interesting and important issue which the search for alternatives in the treatment of colitis.

In general, the manuscript is well written, in my opinion the quality of presentation is high.

I have some minor comments which are as follows:

line 97 - more details about the diet can be provided

Why the authors decided 38 day experimental period?

On which basis 3% DSS was chosen?

Figures should be of better quality.

Table 1- the title is unclear, do the values in the table present % of the weigfht of particular organs relative to whole body weight?

References in the text shoule be in [ ].

Author Response

请参阅附件。

Reviewer 3 Report

The main question addressed by the research is the study focused on the role of WPH in inflammatory processes to elucidate the protective effect of WPH on DSS-induced colitis in mice, as well as the potential mechanisms underlying its action, aiming to provide data to develop and apply new functional ingredients to treat colitis. The manuscript shows points of scientific interest. Well-designed research. The topic is relevant in the scientific, biotechnological, industrial, clinical, and nutritional areas. However, the points indicated below must be reviewed and improved by the Authors, considering their relevance for the visibility of the publication.

1) The authors should deepen the discussion. A greater association is needed between the results and data described in the scientific literature.

2)  The discussion emphasizes the results in many items. These are important but not discussed properly. The citations of articles in the discussion and their correlation with the results of this research are often missing or confusing.

3) Item 3.1 The composition of WPH should be better described. Explore the result and compare it with data described in the literature.

4) Lines 206 e 207. Authors cite a reference without discussion. The phrase seems to indicate that the results were described by ref. [19].

5) Check the discussion on lines 253 to 256. Similar to the previous point, the reference is cited, leading us to think that the results are from ref. [20].

6) Lines 276 to 277 show a situation similar to that described in previous points.

7) Overall, the Authors must improve the discussion to meet the question proposed in the objective.

8) Lines 261 to 264. The possible mechanisms of WPH intervention are an important point that should be better described and deepened.

9) The quality/resolution of the figures should be improved. Many of them are diffuse.

10) Conclusions addressed by the research are consistent with experimental evidence.

11)   Authors should check the Nutrient Guidelines to format the manuscript.

12)  The citation number in the text must be placed in brackets. This makes it easier to find. Please see Nutrients Guidelines.

An English-language review of the manuscript is required. It is indicated in Review Report Form.

Author Response

请参阅附件。
